# Clustering of home delivery in Bangladesh and its predictors: Evidence from the linked household and health facility level survey data

**Kaniz Fatima**[1], **Shimlin Jahan Khanam**[2], **Md Mostafizur Rahman**[1], **Md Iqbal Kabir**[3], **Md Nuruzzaman Khan**[2,4]*

1 Department of Population Science and Human Resource Development, University of Rajshahi, Rajshahi, Bangladesh, 2 Department of Population Science, Jatiya Kabi Kazi Nazrul Islam University, Mymensingh, Bangladesh, 3 Climate Change and Health Promotion Unit, Ministry of Health and Family Welfare, Dhaka, Bangladesh, 4 Faculty of Health and Medicine, University of Newcastle, Newcastle, New South Wales, Australia

* mdnuruzzaman.khan@uon.edu.au

**Data Availability Statement:** The data supporting the findings of this study are accessible through Demographic and Health Survey but are not

## Abstract

Around half of births in Bangladesh occur at home without skilled birth personnel. This study aims to identify the geographical hot spots and cold spots of home delivery in Bangladesh and associated factors. We analyzed data from the 2017/2018 Bangladesh Demographic and Health Survey and the 2017 Bangladesh Health Facility Survey. The outcome variable was home delivery without skilled personnel supervision (yes, no). Explanatory variables included individual, household, community, and healthcare facility level factors. Moran's I was used to determine hot spots (geographic locations with notably high rates of home delivery) and cold spots (geographic areas exhibiting significantly low rates of home delivery) of home delivery. Geographically weighted regression models were used to identify cluster-specific predictors of home delivery. The prevalence of without skilled personnel supervised home delivery was 53.18%. Hot spots of non-supervised and unskilled supervised home delivery were primarily located in Dhaka, Khulna, Rajshahi, and Rangpur divisions. Cold spots of home delivery were mainly located in Mymensingh and Sylhet divisions. Predictors of higher home births in hot spot areas included women's illiteracy, lack of formal job engagement, higher number of children ever born, partner's agriculture occupation, higher community-level illiteracy, and larger distance to the nearest healthcare facility from women's homes. The study findings suggest home delivery is prevalent in Bangladesh. Awareness-building programs should emphasize the importance of skilled and supervised institutional deliveries, particularly among the poor and disadvantaged groups.

## Introduction

Maternal morbidity and mortality are significant public health threats globally, especially in low- and middle-income countries (LMICs) [1, 2]. Each day, approximately 810 women in LMICs die from pregnancy-related causes during and after delivery, most of which are

publicly available. Researchers interested in accessing the dataset can do so by submitting a research proposal to Demographic and Health Survey, similar to the process we followed to obtain the dataset for this study. The dataset can be accessed at https://dhsprogram.com/data/. Interested researchers can apply to access the datasets at https://dhsprogram.com/data/.

**Funding:** The authors received no specific funding for this work.

**Competing interests:** The authors have declared that no competing interests exist.

preventable through adequate delivery healthcare services, including hospitalized and skilled supervised delivery [3, 4]. Recognizing the importance of ensuring delivery healthcare services, policymakers have prioritized this as a key strategy to achieve the Millennium Development Goals' target of reducing maternal mortality by one-third from the 1990's level by 2015 [5]. However, LMICs, including Bangladesh, have not made significant progress in increasing the utilization of delivery healthcare services [6, 7]. Given the ongoing critical importance, there is now an increased focus on ensuing delivery healthcare services, specifically in alignment with the Sustainable Development Goals (SDGs), with a particular emphasis on SDG 3.8. This goal aims to ensure universal access to sexual and reproductive healthcare services, ultimately working towards the target of reducing maternal mortality to 70 deaths per 10,000 live births by 2030. However, in LMICs, the absence of skilled attendants during labor, delivery, and the early postpartum period accounts for 16–33% of total maternal mortality [8]. Consequently, LMICs will be unable to achieve the target of reducing maternal mortality without significant progress in ensuring delivery healthcare services [9].

To achieve universal coverage of delivery healthcare services, it is crucial to understand the dynamic nature of its utilization and the factors influencing it. Assessing the current progress and the situation at the area level is particularly important for LMICs, where women's socio-demographic conditions vary [9]. This highlights the need for area-level analysis to identify determinants and respond accordingly. However, existing studies in LMICs and Bangladesh mainly focus on basic characteristics of women as determinants of home delivery, such as illiteracy, lack of formal employment, and lower wealth quintile [10, 11]. Moreover, available studies provide average estimates for these factors across entire study area, despite their variations across different areas. These variations are influenced by changing community-level norms and perceptions regarding the utilization of delivery healthcare services. Therefore, it is important to obtain area-level estimates of delivery healthcare services utilization. This study aims to address these limitations by exploring the geographical hot spots and cold spots of home delivery in Bangladesh and identifying area-level predictors of home delivery.

## Methods

### Study design and population

We analyzed data from the 2017/2018 Bangladesh Demographic and Health Survey (BDHS) and the 2017 Bangladesh Health Facility Survey (BHFS). These surveys were conducted by the Ministry of Health and Family Welfare in Bangladesh, supervised by the National Institute of Population Research and Training. Further information about these nationally representative cross-sectional surveys has been published elsewhere [12, 13], but a brief description is provided here. The 2017 BDHS gathered data from married women of reproductive age residing permanently or spending the previous night in selected households. Household selection occurred in two stages. Firstly, 675 clusters (an area with 100/120 households) were randomly chosen from a list of 293,579 clusters in Bangladesh, generated based on the 2011 Bangladesh National Population Census. After excluding three flooded clusters, 672 clusters remained. Secondly, 30 households were randomly selected from each cluster. The survey was conducted in 19,457 households, with an inclusion rate exceeding 96%. Among the 20,376 women in these households, 20,127 were interviewed, resulting in a response rate of 98.8%. Further information regarding the sampling procedure can be found elsewhere [12].

The 2017 BHFS randomly selected 1524 healthcare facilities across Bangladesh, encompassing primary, secondary, and tertiary level establishments. From a list of 19,811 registered healthcare facilities, 1600 were initially considered. The selected facilities represented a comprehensive range, including public, private, and non-government institutions.

Both the 2017/18 BDHS and 2017 BHFS collected spatial data, including latitude and longitude coordinates for each surveyed cluster. These coordinates were recorded during data collection using a geographical pointing system.

## Analytical sample

A total of 4,948 women's data were extracted from the main sample with the condition, (i) reported at least one life birth within three years of the survey (ii) reported methods of delivery, and (iii) reported who provided supervision during delivery.

## Study variables

The study variable was home delivery with no or unskilled personnel supervision (Yes or No). The pertinent information was gathered by asking women, "*Where was your most recent delivery*?" and *"Who assisted with the delivery*? ". If women report that their delivery occurred at home without the presence of any skilled personnel, then it was considered as no or unskilled supervised home delivery.

## Explanatory variable

The study examined various explanatory variables at the individual-, household-, community-, and healthcare facility levels, which were selected based on relevant literature from Bangladesh and other LMICs [14, 15]. A comprehensive search was performed in five databases (Pubmed, Cinhal, Embase, Web of Science, and Google Scholar) using specific keywords related to delivery methods. The study examined the associations between these variables and the absence of skilled personnel supervision. Significant variables were selected for analysis.

At the individual level, factors such as women's age at birth, education level, and employment status were considered. Household-level factors included husband's education, husband's occupation, number of children previously born, preceding birth interval, family type, and wealth quintile. Community-level factors comprised of place of residence, division, and community-level illiteracy. The calculation procedure for this community-level factor, which was derived from aggregated education responses by wealth index, can be found in detail elsewhere [16]. Additional variable considered was distance from women's homes to nearest healthcare facility where delivery healthcare services are available. We calculated distance in two stages. During the initial phase, we identified clusters of households where women reside in close proximity to health facilities offering hospital delivery care services. In the subsequent step, we utilized road communication system data from Bangladesh to calculate the on-road distance that later incorporated in the analysis. The calculation procedure for this distance variable is available elsewhere [17].

## Statistical analysis

The study initially computed proportions of non-supervised or unskilled supervised home delivery and explanatory variables across the 672 clusters of the BDHS 2017/18. Survey weights were applied using the svy command in Stata to obtain these proportions. They were then merged with the spatial locations of the BDHS clusters.

To determine the distribution pattern of non-supervised or unskilled supervised home delivery in the study area, spatial autocorrelation (Global Moran's I statistics) was employed. The Getis-Ord Gi* statistic was utilized to assess spatial autocorrelation variation across study locations, calculating Gi* statistics for each area. Z-scores and corresponding p-values were calculated to evaluate the statistical significance of clustering in the sampling units. The study

incorporated a false discovery rate (FDR) correction when using the Getis-Ord Gi* (d) statistics to account for multiple dependent tests. The significance of applying the FDR correction method in DHS data has been discussed elsewhere [17, 18].

A Geographically Weighted Regression (GWR) model was utilized to determine the cluster-specific coefficients of explanatory variables associated with non-supervised or unskilled supervised home delivery. The reason for using GWR was to provide cluster-level estimates for explanatory variables concerning their effects on home delivery that other regression models, including ordinary least square (OLS) regression model, could not provide; they only offer the average estimate for the sample. Variables meeting the GWR model assumptions were selected using exploratory regression and ordinary least square (OLS) regression models, along with their respective tests. The exploratory regression model was employed initially to identify variables for inclusion in the OLS model, which aimed to identify predictors of the observed spatial pattern of non-supervised or unskilled supervised home delivery in Bangladesh. Descriptive statistics were computed using Stata software version 15.1 (Stata Corporation, College Station, Texas, USA). For all other analyses, the statistical package R (version 4.1.1) was utilized.

**Ethics statement.** We analyzed secondary 2017 BDHS data, which were obtained in a deidentified form from the custodian of the Demographic and Health Survey Program. The survey had received ethical approval from the Bangladesh Medical Research Council and the Demographic and Health Survey Program of the USA. Therefore, no additional ethical approval was necessary for conducting this study. Informed consent was obtained from each respondent before including them in the survey.

## Results

### Background characteristics of the respondents

Table 1 presents the socio-demographic characteristics of the 4,948 women included in this study. The majority of respondents (70.63%) were aged 20 to 34 years. Approximately 27.69% had primary education, 48.83% had secondary education, and nearly 17% had higher education. Around 71% of women had 1–2 children, and three-fourths (75.42%) had an interval of four years or more between their two most recent pregnancies. Among the participants, 73.22% resided in urban areas.

**Geographical distribution of non-skilled personnel supervised home delivery prevalence.** Table 2 presents the geographic distribution of non-skilled personnel supervised home delivery prevalence in Bangladesh. This study found a significant difference in the prevalence of non-skilled personnel supervised home delivery across different places of residence and divisions. Urban areas exhibited a higher prevalence (68%) compared to rural areas (48%). Among the eight administrative divisions, Dhaka had the highest prevalence (61%), while Sylhet division had the lowest prevalence (40.73%).

**Hot spots and cold spots of non-supervised and unskilled supervised home delivery according to the Getis-Ord Gi*.** The Getis-Ord Gi* statistic indicated significant clustering (p < 0.01) (Fig 1). Cluster-level spatial analysis (Fig 1) identified significant hot spots (geographic locations with notably high rates of home delivery) of non-supervised and unskilled supervised home delivery in Dhaka, Khulna division, parts of Rajshahi and Rangpur division, as well as areas in Barishal and Chattogram division. Conversely, significant cold spots (geographic areas exhibiting significantly low rates of home delivery) were observed in Sylhet and Mymensingh divisions, along with some parts of Barishal, Rangpur, and Rajshahi divisions.

**Model comparisons—Ordinary least square (OLS) and Geographically Weighted Regression (GWR).** Table 3 displays the OLS model results for non-skilled personnel

**Table 1. Characteristics of respondents, Bangladesh 2017/18 (N = 4,948).**

| Characteristics | Overall %, 95% CI |
|---|---|
| **Women' age at birth** | |
| Median (IQR) | 26.00 (21–31) |
| ≤19 years | 25.14 (23.78–26.55) |
| 20–34 years | 70.63 (69.16–72.06) |
| ≥35 years | 4.23 (3.67–4.86) |
| **Women's education** | |
| Illiterate | 6.33 (5.49–7.28) |
| Primary | 27.69 (25.90–29.55) |
| Secondary | 48.83 (47.0–50.67) |
| Higher | 17.16 (15.68–18.74) |
| **Women's working status** | |
| Yes | 37.20 (35.06–39.40) |
| No | 62.80 (60.60–64.94) |
| **Husband occupation** | |
| Agriculture | 19.44 (17.81–21.17) |
| Physical worker | 53.08 (51.16–55.00) |
| Services or business | 6.91 (5.07–7.65) |
| **Number of children ever born** | |
| Mean number (SE) | 2.72 (1.40) |
| 1–2 children | 71.02 (69.40–72.59) |
| ≥3 children | 28.98 (27.41–30.60) |
| **Preceding birth interval** | |
| < 2 years | 6.0 (5.29–6.80) |
| 2–4 years | 18.58 (17.34–19.89) |
| >4 years | 75.42 (73.95–76.83) |
| **Types of family** | |
| Nuclear | 31.14 (29.45–32.87) |
| Joint | 68.86 (67.13–70.55) |
| **Wealth quintile** | |
| Poorest | 20.63 (18.60–22.83) |
| Poorer | 20.55 (19.04–22.15) |
| Middle | 19.18 (17.66–20.80) |
| Richer | 20.14 (18.40–22.01) |
| Richest | 19.50 (17.58–21.57) |
| **Place of residence** | |
| Rural | 26.78 (25.10–28.52) |
| Urban | 73.22 (71.48–74.90) |
| **Division** | |
| Barishal | 5.7 (5.1–6.3) |
| Chattogram | 21.2 (19.6–23.0) |
| Dhaka | 25.7 (24.0–27.5) |
| Khulna | 9.2 (8.3–10.1) |
| Mymensingh | 8.56 (7.7–9.5) |
| Rajshahi | 11.53 (10.3–12.89) |
| Rangpur | 10.53 (9.49–11.67) |
| Sylhet | 7.55 (6.69–8.59) |
| **Community level illiteracy[+]** | |

(*Continued*)

**Table 1.** (Continued)

| Characteristics | Overall %, 95% CI |
|---|---|
| Low (≤ 20%) | 46.95 (42.41–51.54) |
| Moderate (21.0–49.0) | 42.73 (38.18–47.41) |
| High (≥ 50) | 10.2 (7.66–13.77) |

Note

[+]Low is defined as illiteracy in the community below 25%, moderate as between 25% and 50%, and high as above 50%. [++++]Measured as total fertility rate, with low and high defined as at or below 2.1 and above 2.1, respectively. Notes: All values are weighted percentages unless otherwise noted; percentages may not add to 100% because of rounding.

supervised home delivery in Bangladesh. The model was constructed using exploratory regression analysis, ensuring no multicollinearity among explanatory variables. The AIC values for the OLS and GWR models were 1020.30 and 1136.23, respectively (Table 3, Table 4). The adjusted R-square for GWR (0.78) was 7% higher than OLS (0.71), indicating a better fit for this study.

**Predictors of non-skilled personnel supervised home delivery.** Fig 2A–2C presents the cluster-wise coefficients of the GWR model for individual-level predictors of non-skilled personnel supervised home delivery. Maternal age of 20–34 years (Fig 2B) and working status (Fig 2C) emerged as significant predictors in regions with prevalent hot spots (Rajshahi, Dhaka, and Khulna divisions). In the Sylhet and Mymensingh divisions, where most cold spots were observed, the absence of formal education among women (Fig 2A) was a significant predictor.

Fig 3A–3F displays the cluster-wise coefficients of the GWR model for household-level predictors. In hotspot areas, partner illiteracy, having more children, and the partner's occupation as an agricultural worker were significant predictors. Conversely, in cold spot areas, these characteristics had less impact on increasing home delivery.

Fig 4A–4B illustrates the effects of community and healthcare facility-level factors on home delivery. The study identified community-level illiteracy as a significant predictor of non-skilled personnel supervised home delivery in hot spots areas.

**Table 2. Weighted proportion of non-use of non-skilled personnel supervised home delivery in Bangladesh by place of residence and administrative division, BDHS 2017/18.**

| | Non-skilled personnel supervised home delivery n (%, 95%CI) | P-value |
|---|---|---|
| **Total** | 53.18 (50.79–55.55) | |
| **Place of residence** | | |
| Urban | 68.02 (64.33–71.50) | p<0.001 |
| Rural | 47.75 (44.85–50.65) | |
| **Administrative Division** | | |
| Barishal | 47.28 (40.99–5365) | p<0.001 |
| Chattogram | 51.49 (45.38–57.55) | |
| Dhaka | 60.58 (54.92–65.96) | |
| Khulna | 6.81 (58.54–68.77) | |
| Mymensingh | 41.64 (36.57–46.89) | |
| Rajshahi | 54.51 (49.07–59.84) | |
| Rangpur | 49.25 (41.82–56.72) | |
| Sylhet | 40.73 (33.73–48.12) | |

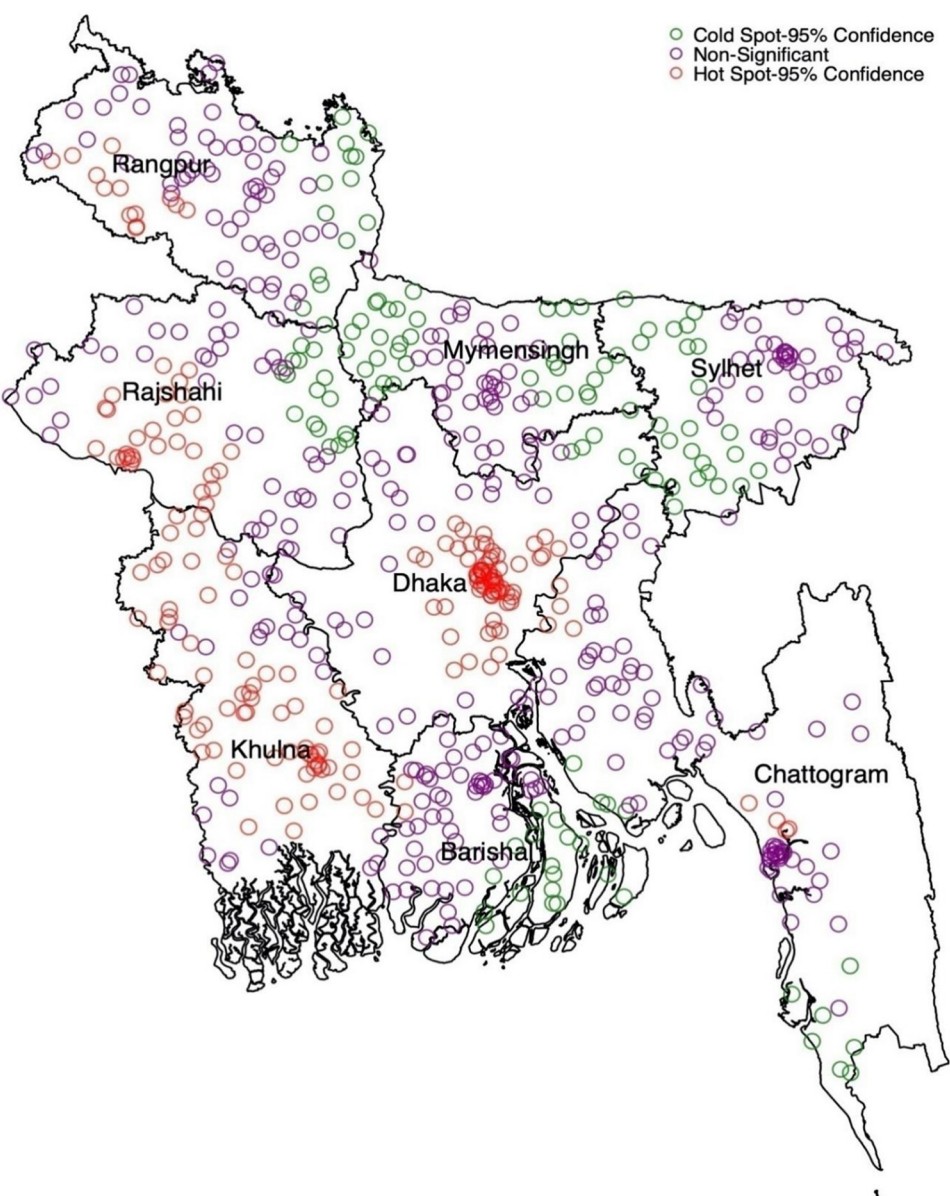

**Fig 1. Hot spot and cold spot of non-supervised and unskilled supervised home delivery in Bangladesh (we generated this map using data obtained from the survey, with the base shapefile sourced from freely available source: https://data.humdata.org/dataset/cod-ab-bgd?).**

## Discussion

This study aimed to identify geographical hot spots and cold spots of non-skilled personnel supervised home births in Bangladesh and their associated factors. Hot spots were mainly located in Rajshahi, Rangpur, and Dhaka divisions, while cold spots were observed in Sylhet and Mymensingh divisions. Among individual-level factors, women aged 20–34 years and those without formal employment had a significant influence in the hot spots. At the household level, joint family status had a negative impact on home delivery. However, the poorest wealth quintile and partner's agricultural occupation contributed to increased home delivery. Higher community illiteracy and longer distance to healthcare facilities were also predictors of increased home delivery in hot spots.

**Table 3. Ordinary least square regression model identifying significant factors of non-skilled personnel supervised home delivery in Bangladesh, BDHS 2017/18.**

| Variable category | Coefficient | Standard error | t-statistics | Probability | Robust std-error | Robust t-statistics | Robust probability | VIF |
|---|---|---|---|---|---|---|---|---|
| Illiterate women | 0.08 | 0.03 | 42.0 | <0.01 | 14.0 | 13.2 | <0.01 | 3.12 |
| Women's age 20–34 | 0.18 | 0.09 | 10.0 | <0.01 | 8.0 | 13.2 | <0.01 | 2.12 |
| Women's engagement in formal occupation | 0.14 | 0.04 | 39.0 | <0.01 | 24.0 | 16.9 | <0.01 | 2.30 |
| Women's husband was illiterate | 0.09 | 0.02 | 20.7 | <0.01 | 12.0 | 08.2 | <0.01 | 4.20 |
| Preceding birth interval | 0.12 | 0.03 | 18.0 | <0.01 | 11.8 | 12.4 | <0.01 | 3.80 |
| Women's husband was agricultural worker | 0.09 | 0.02 | 40.0 | <0.01 | 20.0 | 60.3 | <0.01 | 2.39 |
| Number of children ever born | 0.04 | 0.01 | 19.7 | <0.01 | 08.7 | 13.0 | <0.01 | 2.20 |
| Joint family types | 0.07 | 0.01 | 27.0 | <0.01 | 21.0 | 9.9 | <0.01 | 3.18 |
| Poorest wealth quintile | 0.17 | 0.02 | 12.6 | <0.01 | 13.3 | 17.0 | <0.01 | 1.87 |
| Moderate community level illiteracy | 0.14 | 0.03 | 19.0 | <0.01 | 09.6 | 14.1 | <0.01 | 1.85 |
| Average distance of nearest healthcare facility | 0.10 | 0.01 | 22.9 | <0.01 | 21.1 | 40.4 | <0.01 | 1.90 |
| **Model diagnostics** | | | | | | | | |
| Number of observation (EAs) | 672 | | Akaike's Information Criterion (AIC): | | | | 1020.30 | |
| Multiple R-squire | 0.80 | | Adjusted R-square | | | | 0.71 | |
| Joint F-Statistics | 1492.13 | | Probability (> F), (11,669) degrees: | | | | <0.01 | |
| Joint Wald Statistics | 152.23 | | Probability (> chi-squared) | | | | <0.01 | |
| Koenker (BP) Statistics | 1662.14 | | Probability (> chi-squared) | | | | <0.01 | |
| Jarque-Bera Statistics | 0.66 | | Probability (> chi-squared) | | | | <0.01 | |

The observed prevalence of non-skilled personnel-supervised home delivery is 53.18, aligning with the findings of another study in Bangladesh [19]. However, this rate is significantly higher than the average in LMICs, where a recent study reported that 30% of total births occur at home [20]. Instead of focusing primarily on international and government-level efforts to ensure universal coverage of delivery care services such higher prevalence of unskilled supervised home delivery is a matter of concern. This is likely due to a combination of failures at the healthcare facility level and a lack of awareness among respondents regarding the importance of accessing delivery healthcare services from skilled healthcare personnel.

**Table 4. Geographically weighted regression model assessing factors of no or unskilled supervised home delivery in Bangladesh, BDHS 2017/18.**

| GWR model parameters | GWR model statistics |
|---|---|
| Residual squares | 16.33 |
| Effective number | 60.00 |
| Sigma | 0.24 |
| AIC | 1136.23 |
| Multiple R-square | 0.80 |
| Adjusted R-square | 0.78 |

Note: Explanatory variables considered in the GWR model were Illiterate women, women aged 20–34 years, women's no formal job engagement, illiteracy of the women's partner, number of children ever born, four years or more preceding birth interval, women's joint family status, women's poorest households wealth quintile, women's partner was agricultural worker, community level higher illiteracy, and average distance of women's homes from nearest healthcare facility.

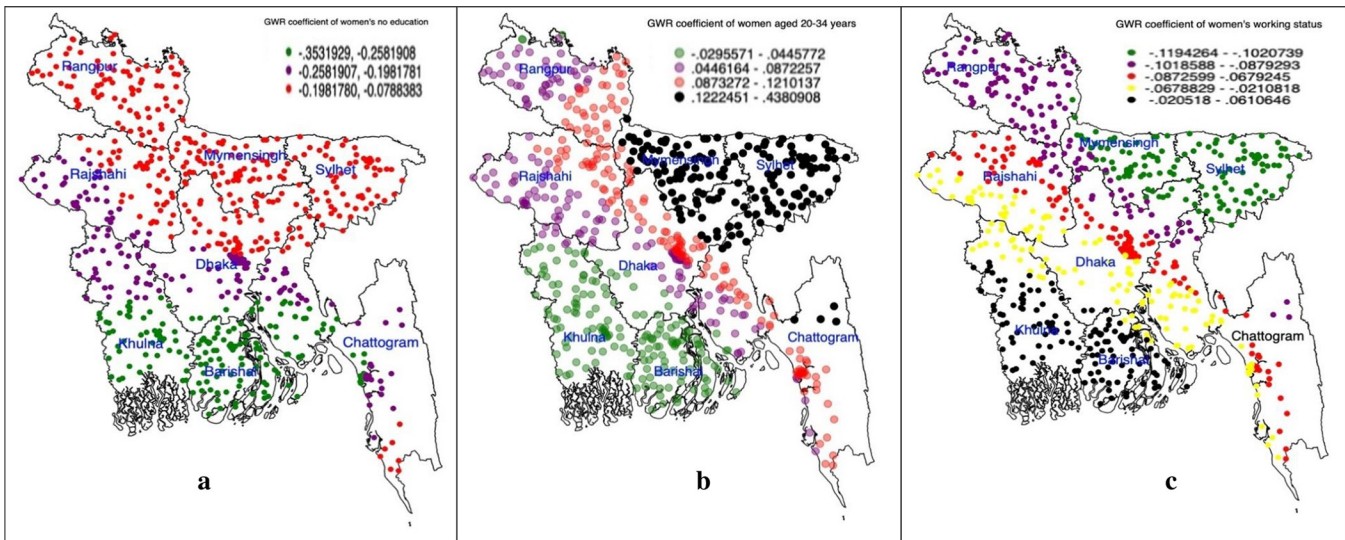

**Fig 2.** (a-c): Individual level predictors of non-skilled personnel supervised home delivery accessed through using the geographically weighted regression model (we generated these maps using data obtained from the survey, with the base shapefile sourced from freely available source: https://data.humdata.org/dataset/cod-ab-bgd?).

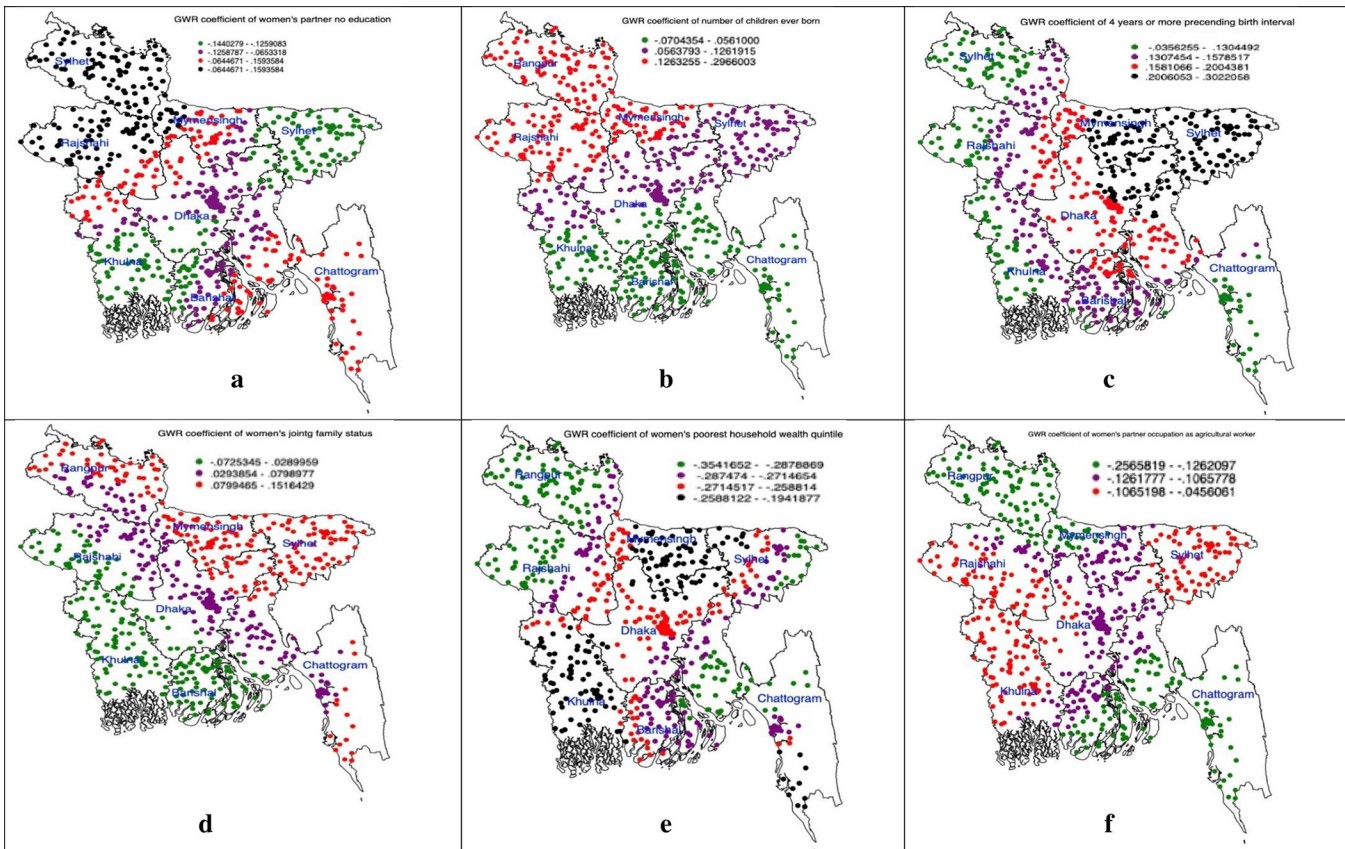

**Fig 3.** (a-f): Household level predictors of non-skilled personnel supervised home delivery accessed through using the geographically weighted regression model (we generated these maps using data obtained from the survey, with the base shapefile sourced from freely available source: https://data.humdata.org/dataset/cod-ab-bgd?).

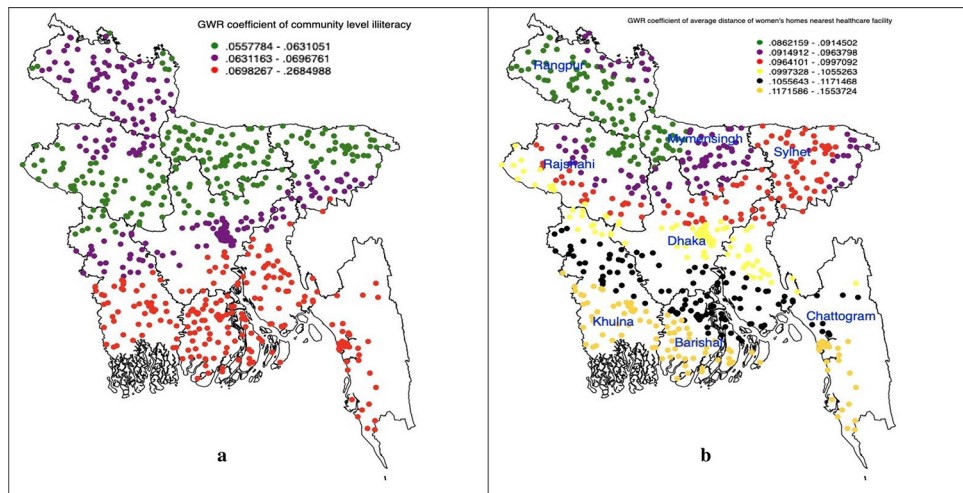

**Fig 4.** (a-b): Community level predictors of non-skilled personnel supervised home delivery accessed through using the geographically weighted regression model (we generated these maps using data obtained from the survey, with the base shapefile sourced from freely available source: https://data.humdata.org/dataset/cod-ab-bgd?).

Hot spots of home births in Bangladesh are primarily located in the Rajshahi, Rangpur, and Dhaka divisions, which contradicts available evidence indicating higher rates in the Sylhet, Barisal, and Chattogram divisions [21]. This disparity may be attributed to differences in the estimation process. Our analysis focused on estimating the prevalence of home births across clusters to clearly identify cluster-level differences. However, existing studies provided average estimates for all clusters within specific divisions, often masking clusters with high levels of home births. The key reasons for the clustering of home births are numerous, but the most significant factors are likely the uneven distribution and distance of healthcare facilities, as well as the availability of healthcare personnel, which emerged as dominant factors in the hot spots [22, 23]. These effects may be even more pronounced for women with lower education, no job engagement, and a higher number of children, or possibly a combination of these factors. Illiteracy reduces women's knowledge about the importance of skilled personnel supervised delivery [22, 23]. Lack of job engagement diminishes women's empowerment and increases their dependence on their husbands to access healthcare services and cover delivery costs [24]. A higher number of children ever born increases women's inclination to rely on their past delivery experiences [25]. The age of women is also an important factor to consider. Hospitalized deliveries are often prioritized for women of comparatively younger and older ages, considering their higher risk of facing pregnancy complications. Therefore, healthcare providers at the community level often recommend hospitalized delivery for them as compared to younger women [4, 22]. These challenges often stem from community-level norms and traditions related to the use of delivery healthcare services. For instance, visiting a healthcare facility during delivery may expose women to the risk of evil attacks [22]. Additionally, healthcare facilities in Bangladesh are still predominantly staffed by male healthcare personnel, leading to situations where women may need to seek delivery healthcare services from male providers. At the community level, the likelihood of such risks is widely acknowledged as a significant concern. As a result, a significant portion of women do not access delivery healthcare services due to their fear of receiving care from male personnel [2, 26, 27]. These issues are particularly relevant in communities with higher illiteracy rates and in the hot spots of home birth prevalence [23].

Cold spots of home births are primarily located in the Sylhet and Mymensingh divisions. This observation contradicts available studies, but it can be attributed to variations in the estimation procedures mentioned earlier. An interesting finding of this study is the significant role played by increasing birth intervals and the poorest wealth quintile in promoting skilled supervised delivery. These findings are novel and require further explanation. Rising preceding birth intervals can enhance delivery healthcare services in multiple ways. For example, longer birth intervals raise awareness among women about possible complications and the need for extra care for the upcoming child. Furthermore, women with a longer duration between pregnancies do not have a very young child at home requiring constant care, making it easier for them to decide on hospitalized delivery. Previous studies in Bangladesh have also reported longer birth intervals among wealthier and more educated women, as well as those who use modern contraception [28, 29]. These connections highlight the rising awareness of the importance of delivery healthcare services through formal education and interactions with family healthcare providers during family planning and contraception discussions. However, the reasons for higher access to delivery healthcare services among women in the poorest wealth quintile remain unknown. One potential explanation could be that healthcare services in these cold spot areas are functioning well, particularly for the poorest segment of the population. Further studies are recommended to explore this issue more comprehensively.

This study highlights significant regional variations in non-skilled personnel supervised home delivery in Bangladesh, emphasizing the need for area-specific policies and programs. Promoting women's education and reducing the distance to healthcare facilities are crucial for reducing home delivery across all areas. Raising awareness among women and ensuring accessible healthcare services at the community level are essential. Renovating existing community clinics can enhance the provision of delivery healthcare services.

The study's strength lies in its unique combination of household and healthcare facility data, providing conclusive findings for the entire country. It offers valuable insights into geographical variations and cluster-level predictors, empowering policymakers to develop more targeted policies and programs. However, this study has some limitations that should be considered. Firstly, the use of cross-sectional data may introduce recall bias in the outcome and exposure variables. To mitigate this, the BDHS implemented measures such as follow-up questions and collection of delivery care data within a specific time frame. However, recall bias cannot be completely eliminated. Additionally, the study did not incorporate the influence of social norms and awareness due to data limitations. Furthermore, cluster locations in the figures were displaced for respondent confidentiality, although they closely represent the actual locations. Despite these limitations, the findings of this study are valuable for policymakers and program makers in shaping policies and programs to enhance the utilization of delivery healthcare services.

## Conclusion

The study revealed significant geographical variations in non-skilled personnel supervised home births in Bangladesh. Hot spots of such births were predominantly located in the Dhaka, Khulna, and Rajshahi divisions, while cold spots were primarily found in Sylhet and Mymensingh divisions. The factors associated with hot spots and cold spots varied across different clusters in Bangladesh. However, increasing education and reducing the distance to the nearest healthcare facilities were consistently important factors for promoting the utilization of delivery healthcare services among women. This underscores the need for localized policies and programs to enhance the use of such services, rather than relying solely on national-level approaches. An awareness-raising campaign should be implemented nationwide to emphasize

the importance of accessing healthcare services during delivery, irrespective of whether the area is classified as a hot spot or a cold spot.

## Acknowledgments

The authors thank the MEASURE DHS for granting access to the 2017/18 BDHS data.

## Author Contributions

**Conceptualization:** Kaniz Fatima, Shimlin Jahan Khanam, Md Mostafizur Rahman, Md Nuruzzaman Khan.

**Data curation:** Kaniz Fatima, Md Mostafizur Rahman.

**Formal analysis:** Kaniz Fatima, Shimlin Jahan Khanam, Md Mostafizur Rahman, Md Nuruzzaman Khan.

**Software:** Kaniz Fatima.

**Supervision:** Kaniz Fatima, Md Mostafizur Rahman.

**Validation:** Md Mostafizur Rahman.

**Writing – original draft:** Kaniz Fatima, Shimlin Jahan Khanam, Md Mostafizur Rahman, Md Nuruzzaman Khan.

**Writing – review & editing:** Kaniz Fatima, Shimlin Jahan Khanam, Md Mostafizur Rahman, Md Iqbal Kabir, Md Nuruzzaman Khan.

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
