## [Decision Letter · Decision Letter 0]

27 Dec 2023

PGPH-D-23-02112

Clustering of home delivery in Bangladesh and its predictors: evidence from the linked household and health facility level survey data

Dear Dr Khan

Thank you for submitting your manuscript to PLOS Global Public Health. After careful consideration, we feel that it has merit but does not fully meet PLOS Global Public Health’s publication criteria as it currently stands. Therefore, we invite you to submit a revised version of the manuscript that addresses the points raised during the review process.

We look forward to receiving your revised manuscript.

Kind regards,

Gizachew Tessema, PhD

Academic Editor

Journal Requirements:

1. We would like to request copy editing.

Additional Editor Comments (if provided):

Reviewers' comments:

Reviewer's Responses to Questions

**Comments to the Author**

1. Does this manuscript meet PLOS Global Public Health’s publication criteria? Is the manuscript technically sound, and do the data support the conclusions? The manuscript must describe methodologically and ethically rigorous research with conclusions that are appropriately drawn based on the data presented.

Reviewer #1: Partly

Reviewer #2: Yes

2. Has the statistical analysis been performed appropriately and rigorously?

Reviewer #1: I don't know

Reviewer #2: I don't know

3. Have the authors made all data underlying the findings in their manuscript fully available (please refer to the Data Availability Statement at the start of the manuscript PDF file)?

Reviewer #1: No

Reviewer #2: No

4. Is the manuscript presented in an intelligible fashion and written in standard English?

Reviewer #1: Yes

Reviewer #2: Yes

5. Review Comments to the Author

Reviewer #1: Thank you very much for the opportunity to review this well-written and interesting manuscript. The authors tackle an important question – predictors of home birth in Bangladesh – using a novel approach. They make an important contribution to the literature by showing substantial geographic variation in home births and identifying important predictors of home birth.

Please see my comments below.

Methods:

1) What did you do for cases in which a single woman had had 2 or more births in the past 3 years? Did you just use the most recent, or did you use all of them?

2) It would be very helpful if you could include clearer definitions of “hot spots” and “cold spots.”

3) In addition, I think the current use of the term “clusters” is somewhat confusing. I think you might be using it in two separate ways: the first usage is to define the DHS clusters (used for sampling), and the second is to define geographic groupings of home births (as in “prevalence of home birth clusters”). I would recommend clarifying the usage of this term throughout the manuscript.

Results:

1) In the justification for using GWR over OLS, it would be useful to hear your reflections on whether and why GWR might be the more appropriate way to model these data (based on theory about the relationships being modelled, as opposed to R-squared and AIC).

2) Figures 2-4 – I found the discussion of these results very interesting, but the figures somewhat hard to understand. It would be helpful if you made the legends larger. Additionally, I’m not sure I understand the coefficients. Why are ranges given instead of point estimates with confidence intervals? Is this because the coefficients are estimated separately within every DHS cluster? Finally, would it be possible to use colors in a way that aligns with the numerical values (e.g., one color for each panel, but different color densities to indicate higher vs lower coefficients)?

3) “In hotspot areas, partner illiteracy, having more children, and the partner's occupation as an agricultural worker were significant predictors. Conversely, in cold spot areas, these characteristics had less impact on reducing home delivery.”…Should this instead say, “Conversely, in cold spot areas, these characteristics had less impact on increasing home delivery.”?

4) “The study identified community-level illiteracy as a significant predictor of non-skilled personnel supervised home delivery.” – Was this true in both hot spots and cold spots?

5) In the results from the GWG model, the authors discuss results from hot spots and cold spots. Are they able to also comment on the regression results in the ‘insignificant’ spots, or is that not feasible with this method?

Discussion:

1) The authors’ statement on the prevalence of nonskilled personal supervised home birth in LMICs (54.8 to 70.8) seems very high. For example, this recent study of DHS surveys says that 1 in 3 women in LMICs delivers at home "Prevalence of home birth among 880,345 women in 67 low- and middle-income countries: A meta-analysis of Demographic and Health Surveys" - PMC (nih.gov). What is driving the difference across these different estimates?

2) The authors provide an interesting analysis of the reasons for clustering of home births. In addition to the reasons discussed, I was interested by the finding that women aged 20-34 are less likely to have a home birth. I wonder if this is an indicator the health system is doing a good job of identifying women outside this age range (<20 and 35+) as higher-risk and successfully steering them to health facility-based childbirth.

3) In the discussion about the role of longer birth intervals, the authors could also mention the fact that, when women wait longer between pregnancies, they no longer have a very young child at home in need of their constant care. This might make facility-based childbirth more feasible for the second child.

4) “Our analysis focused on estimating the prevalence of home birth clusters, while existing studies provided average estimates for all clusters within specific divisions.” – Can you explain more about what this means and why the results are different? I think it would be helpful for many readers if you included a clear definition of “prevalence of home birth clusters” (vs hotspots) early in the paper.

Reviewer #2: The authors present an interesting article on a topic of growing importance. With the limited published literature on this specific topic within the LMICs, their analysis is a valuable addition to the field.

I have few comments:

1. Under the Explanatory variables: 'The calculation procedure for this distance variable is available elsewhere [17]' While it is okay to indicate the references of some procedures in other papers, like this distance variable calculation, a summary of the methodology/procedure should be highlighted in the paper. please take note in all the references like such in the paper.

2. In table 1. under Preceding birth interval. Please check '2 years' and specify which group it belong. it is apparently in the two groups.

3. Be consistent in your decimals. some p values are in 2 decimals and some in 3 decimals. Please be consistent across the paper.

4. There are some minor edits in the text needed to be done.

6. PLOS authors have the option to publish the peer review history of their article (what does this mean?). If published, this will include your full peer review and any attached files.

**Do you want your identity to be public for this peer review?** For information about this choice, including consent withdrawal, please see our Privacy Policy.

Reviewer #1: No

Reviewer #2: No

---

## [Editor Report · Decision Letter 1]

30 Jan 2024

Clustering of home delivery in Bangladesh and its predictors: evidence from the linked household and health facility level survey data

PGPH-D-23-02112R1

Dear Dr. Md. Nuruzzaman Khan

We are pleased to inform you that your manuscript 'Clustering of home delivery in Bangladesh and its predictors: evidence from the linked household and health facility level survey data' has been provisionally accepted for publication in PLOS Global Public Health.

Best regards,

Gizachew Tessema, PhD

Academic Editor